# The POU Transcription Factor POU-M2 Regulates *Vitellogenin Receptor* Gene Expression in the Silkworm, *Bombyx mori*

**DOI:** 10.3390/genes11040394

**Published:** 2020-04-06

**Authors:** Guanwang Shen, Enxiang Chen, Xiaocun Ji, Lina Liu, Jianqiu Liu, Xiaoting Hua, Dan Li, Yingdan Xiao, Qingyou Xia

**Affiliations:** 1Biological Science Research Center, Southwest University, Chongqing 400716, China; gwshen@swu.edu.cn (G.S.); 18696612451@163.com (E.C.); na281812082@163.com (L.L.); jianqiu256@outlook.com (J.L.); huaxiaotingswu@126.com (X.H.); lidan2017@email.swu.edu.cn (D.L.); XYD2069472847@163.com (Y.X.); 2State Key Laboratory of Silkworm Genome Biology, Southwest University, Chongqing 400716, China; 3Chongqing Key Laboratory of Sericulture Science, Chongqing Engineering and Technology Research Center for Novel Silk Materials, Chongqing 400716, China; 4Research Center of Bioenergy & Bioremediation, College of Resources and Environment, Southwest University, Chongqing 400716, China; jicun1221@126.com

**Keywords:** POU transcription factor, *Vitellogenin receptor*, transcriptional regulation, Oogenesis, *Bombyx mori*

## Abstract

Vitellogenin receptors (VgRs) play critical roles in egg formation by transporting vitellogenin (Vg) into oocytes in insects. Although the function of VgR in insects is well studied, the transcriptional regulation of this gene is still unclear. Here, we cloned the promoter of the *VgR* gene from *Bombyx mori* (*BmVgR*), and predicted many POU cis-response elements (CREs) in its promoter. Electrophoretic mobility shift and chromatin immunoprecipitation assays showed that the POU transcription factor POU-M2 bound directly to the CREs of the promoter. Overexpression of POU-M2 in an ovarian cell line (BmNs) enhanced *BmVgR* transcription and promoter activity detected by quantitative reverse transcription PCR and luciferase reporter assays. Analyses of expression patterns indicated that *POU-M2* was expressed in ovary at day two of wandering stage initially, followed by *BmVgR*. RNA interference of *POU-M2* significantly reduced the transcription of *BmVgR* in ovary and egg-laying rate. Our results suggest a novel function for the POU factor in silkworm oogenesis by its involvement in *BmVgR* regulation and expands the understanding of POU factors in insect *VgR* expression.

## 1. Introduction

Vitellogenin (Vg) is one of the main nutrients for oviparous animals in the embryonic stage [1,2]. The vitellogenin receptor (VgR), which is responsible for transferring Vg from the hemolymph to the ovary by endocytosis, provides nutrients for oocyte maturation and ensures embryonic development [3]. The modular structure and function of VgRs are conserved and their functions are well studied among oviparous animals, including insects. However, the transcriptional regulation of *VgR* gene expression has not been well studied.

So far, transcriptional regulation of *VgR* has been reported in only a few species. In 2004, the juvenile hormone (JH) analogue was reported to upregulate the expression of *VgR* in *Solenopsis invicta in vitro* [4]. In 2006, binding motifs for several transcription factors (GATA transcription factor, hepatocyte nuclear factor3/forkhead transcription factor, E74, and Broad Complex) were identified in the upstream region of *Aedes aegypti VgR* (*AaVgR*), and a 1.5-Kb region upstream of its transcription start site was reported to be sufficient to drive tissue- and stage-specific expression of a reporter gene in transgenic animals [5]. In 2012 and 2014, insulin was found to upregulate the *VgR* transcript in the ovary, and an estrogen receptor subtype (Esr1) was found to inhibit the expression of *VgR* in the presence of 17β-estradiol (E2) in *Micropterus salmoides* [6,7]. In 2015 and 2019, the juvenile hormone was reported to regulate the transcription of the *VgR* in *Nilaparvata lugens* [8] and *Colaphellus bowringi* [9]. These studies provide valuable information on the regulation of *VgR*, but also suggest that its regulatory mechanism varies between species. *Bombyx mori* is a well-studied model insect [10,11,12,13,14], and oogenesis and choriogenesis have been examined in some detail [15,16]. Recently, two microRNAs have been reported to involve in regulating the tissue- and stage-specific expression of VgR protein in *B. mori* (BmVgR) [17]. However, there are no reports about the transcriptional regulation of *Bombyx mori* (*BmVgR*) gene. In addition, since VgR has a highly conserved modular structure and is critical for insect reproduction, it is of interest to investigate whether any conserved factors are involved in its regulation.

The POU domain transcription factor is found to be highly conserved throughout the Bilateria [18]. In 1988, pituitary-specific transcription factor 1 (Pit-1) [19], octamer-binding protein 1 (Oct-1) [20], Oct-2 [21] in mammals, and *unc-86* in *Caenorhabditis elegans* (*C. elegans*), were first identified and named as POU domain transcription factors [22]. POU transcription factors consist of a conserved POU-specific domain (POU_S_), and a conserved homeobox domain (POU_H_). Generally, POUs binds to an ATGC sequence, while POU_H_ associates with an A/T-rich sequence [23]. In the past three decades, numerous POU domain transcription factors have been identified. Functional studies showed that these factors play important roles in various physiological processes, including development, metabolism, and immunity [23]. Ventral veins lacking (Vvl) is a homolog of the POU domain transcription factor in the holometabolous insect [24]. To date, only two POU homeodomain transcription factors (POU-M1 and POU-M2) have been identified in *B. mori*. POU-M1 was initially cloned from the silk gland and binds to the octamer element of the *sericin 1* gene, regulating its expression [25,26]. Another transcription factor, POU-M2, differs from POU-M1 by only a few amino acids and has multiple functions. POU-M2 regulates the *diapause hormone and pheromone biosynthesis-activating neuropeptide* (*DH-PBAN*) [27,28,29]. The wing disc epidermal protein (WCP), which plays a key role during morphological variation, has also been reported to be regulated by POU-M2 [27]. POU-M2 was found to participate in the regulation of *steroidogenic enzyme* genes, controlling insect ecdysteroidogenesis [30]. Recently, Lin *et al.* have found that POU-M2 is also involved in the regulation of the *B. mori vitellogenin* gene (*BmVg*) [31]. However, it is unclear whether a POU transcription factor is involved in the regulation of *VgR* transcription in *B.mori*, or even in other insect species.

Here, we found that many POU transcription factor cis-response elements (POU CREs) were located in the upstream regulatory region of *VgR* of six insect species. Then, in *B. mori*, we verified that a POU transcription factor, POU-M2, directly bound to these POU CREs, regulating *BmVgR* expression.

## 2. Materials and Methods 

### 2.1. Animal Strains

The *B. mori* strain Dazao was maintained at the Silkworm Gene Bank in Southwest University (Chongqing, China). Fertilized eggs were incubated at 25 °C and 85% relative humidity for hatching. Larvae were fed with fresh mulberry leaves (*Morus sp.*) at 25 °C, with a photoperiod of 12 h light/12 h dark and 75% relative humidity. Ovary samples were dissected from day 6 of the 5th larval instar (L5D6) to pupal day 3 (P4D) for RNA isolation and transcription analysis.

### 2.2. BmVgR Promoter Isolation and Bioinformatics Analysis

A 2.5 kb upstream region of the *BmVgR* translation initiation site was obtained from the silkworm genome database [32] as a potential promoter sequence. Genomic DNA was extracted from silkworms using a Mini BEST universal Genomic DNA Extraction Kit Ver. 5.0 (TaKaRa, Kyoto, Japan). The primers for genome PCR are listed in Appendix A. The following program was used for PCR: 98 °C for 2 min, 30 cycles of 98 °C for 10 s, 56 °C for 10 s, 72 °C for 15 s, and 72 °C for 2 min. All PCR products were separated on 1.5% agarose. The visualized bands were excised and purified using a Gel Extraction Kit (OMEGA, Guangzhou, China). The PCR products were cloned into the pEAZY-T5 Zero cloning vector (TransGen Biotech, Beijing, China), and their DNA sequences were sequenced at Beijing Genomics Institution (BGI) (Beijing, China). Validated sequences for the upstream region of *BmVgR* were analyzed for the presence of transcription factor-binding sites using the MatInspector program [33] and JASPAR ^2020^ [34].

### 2.3. Vector Construction

The entire 2.5-kb *BmVgR* promoter fragment was inserted into the pGL3-basic luciferase vector (Promega, Madison, WI, USA) (named p-VgRP-2.5k) for analysis of the promoter activity analysis. Three deletion and mutated constructs of the *BmVgR* promoter reporter vectors were produced, i.e., p-VgRP-2.0k, p-VgRP-0.7k, p-VgRP-0.1k, p-VgRP-0.1kM. Mutation at predicted POU CREs in the *BmVgR* 0.1 kb promoter were synthesized by TSINGKE Biological Technology, Beijing, China.

The open reading frames (ORFs) of the *POU-M2* and *EGFP* genes were cloned into the psl 1180-Hr3-A4 SV40 expression vector (stored in our laboratory) for overexpression assays (vectors were named 1180-POU-M2, and 1180-EGFP). Primers for the *BmVgR* promoter and *POU-M2* amplification are shown in Appendix A. All constructs were verified by sequencing.

### 2.4. Cell Culture and Dual Luciferase Reporter (DLR) Assay

The BmNs (corresponding to BmN-SWU1) cell line derived from the ovary of *B. mori* (stored in our laboratory) was cultured at 27 °C in IPL-41 medium (AppliChem, Gatersleben, Germany) containing a 10% FBS (Gibco, Grand Island, NE, USA). Transfection of overexpression and luciferase reporter vectors was performed using the X-treme GENE HP DNA transfection reagent (Roche Applied Science, Basel, Switzerland). Luciferase activity was measured using commercially available kits (Promega, Madison, WI, USA) according to the manufacturer’s instructions. The vector p-IE1-Rlucp (stored in our laboratory) was used to drive the expression of Renilla luciferase via the *B. mori* nuclear polyhedrosis virus *IE1* promoter as an internal control. The vector ratio (w/w) between p-VgRP-2.5k and p-IE1-Rlucp was 1:0.1. The ratio (w/w) between the reporter vector and overexpression vectors was 1:1. The measurement of luciferase activity was performed with the Dual Luciferase Reporter Assay kit (Promega, Madison, WI, USA), using the GloMax-Multi Detection System Photometer (Promega, Madison, WI, USA).

### 2.5. Quantitative Reverse Transcription PCR (RT-PCR) and RT-PCR

Total RNA was extracted using Trizol (Invitrogen, Carlsbad, CA, USA), and genomic DNA contamination was removed by RNase-free DNase I (TaKaRa, Kyoto, Japan). The TransScript® One-Step gDNA Removal and cDNA Synthesis Super Mix (Transgen Biotech, Beijing, China) was used for constructing the cDNA library, according to the manufacturer’s instructions. qRT-PCR was performed using SYBR Premix ExTaq II (Takara, Kyoto, China) on a ABI7500 system (ABI, Carlsbad, CA, USA). The following program was used for the amplification reaction: 95 °C for 3 min, followed by 40 cycles of 95 °C for 3 s, 60 °C for 30 s. The silkworm *eukaryotic translation initiation factor 4A* (*BmTIF4A*) gene was used as an endogenous control [35,36]. The relative mRNA levels of target genes were calculated using the 2^-△△C (T)^ method [37]. Three independent replicates were used for each data set. The primers for qRT-PCR were designed using Primer Premier 6 software (Appendix A).

### 2.6. Electrophoretic Mobility Shift Assay (EMSA)

Electrophoretic mobility shift assay (EMSA) was performed using the Chemiluminescent EMSA Kit (Beyotime, Beijing, China), following the manufacture’s protocol. In brief, DNA target probes including the POU elements were designed and labeled with biotin (synthesized by Sangon Biotech, Shanghai, China). Probes were annealed by incubating the complementary strands at 100 °C for 10 min, followed by slow cooling to 25 °C. To perform the EMSA, 2.5 μM of the annealed labeled probes were incubated with the supplied buffer, the purified POU-M2 protein (as described in [38]), and the nucleoproteins of POU-M2-overexpressing BmNs cells were extracted using a Nuclear and Cytoplasmic Protein Extraction Kit (Beyotime, Beijing, China). Some treatments were also performed with a 50 μM unlabeled competitor probe, a 2.5 μM mutational probe and a rabbit polyclonal antibody against POU-M2 (anti-POU-M2) (0.85 μg/μL). After 20 min at 25 °C, the samples were separated on a 5% acrylamide gel at 100 V for 1 h. Subsequently, the samples were transferred to a nylon membrane (Thermo, Waltham, MA, USA) in 0.5× TBE buffer (45 mM Tris borate and 1 mM EDTA, pH 8.3), then the nylon membrane was cross-linked by ultraviolet light (120 mJ/cm^2^). Migration of the biotin-labeled probe was detected by chemiluminescence, using the Super Signal West Femto Maximum Sensitivity Substrate (Thermo, Waltham, MA, USA).

### 2.7. Chromatin Immunoprecipitation (ChIP)

To confirm the binding of transcription factors to CREs, ChIP assays were performed with the EZ-ChIP kit (Millipore, Boston, MA, USA), according to the manufacturer’s instructions. Briefly, BmNs cells transfected with the overexpression vectors and control BmNs cells were fixed with 37% formaldehyde to cross-link chromatin, and then sonicated to shear the cross-linked chromatin into fragments of 200-1000 bp in length. Immunoprecipitation assays were performed using mouse monoclonal antibody against Myc (anti-Myc) (Beyotime, Beijing, China), anti-POU-M2, or nonspecific mouse/rabbit IgG antibodies. The DNA purified from the immunoprecipitated chromatin was used as a template for PCR amplification. The primers used to amplify the specific region near the potential POU-M2 CRE, as well as nonspecific regions, are listed in Appendix A. The PCR products were separated by electrophoresis in 2% agarose gels and sequenced by the Beijing Genomics Institute (BGI) (Beijing, China).

### 2.8. RNA Interference (RNAi) 

siRNAs targeting POU-M2 (nucleotide positions 441–459) and negative-control siRNA (*EGFP* gene) were designed and synthesized by GenePharma (Shanghai, China) (Appendix A). The siRNA injection time points were established based on the expression profiles of *POU-M2* and *BmVgR*. Injections were performed on days 0/1 of the wandering stage (W0D and W1D). Five micrograms siRNA for each gene were injected into the abdominal cavity of female larvae/prepupae with a glass needle. We used 30 larvae for each gene, and the ovaries were dissected on pupal day 3 (P3D) to verify gene expression. After eclosion, all injected female moths were mated with wild-type males, and the egg-laying rate was determined for statistical analysis.

### 2.9. Statistical Analysis

Statistical values are shown as means ± SEM. Mean values were compared using the Student’s *t* test with the following significance thresholds: * *p* < 0.05, ** *p* < 0.01 and *** *p* < 0.001. Statistical analyses were performed using Graph Pad Prism 6.0 (Graph Pad software, New York, NY, USA). Figures were assembled using Adobe Photoshop CS5 (Adobe, SAN Jose, CA, USA) and Adobe Illustrator CS5 (Adobe, USA).

## 3. Results

### 3.1. Cloning and Analysis of the BmVgR Gene Promoter

To investigate the transcriptional regulation of the *BmVgR* gene, we cloned a 2.5 kb sequence from its upstream regulatory region (Figure 1A). Subsequently, we used this sequence to drive the expression of a firefly luciferase reporter gene (*Luc*) by a dual luciferase reporter assay (DLR assay) (Figure 1B,C). The results suggested this 2.5 kb sequence could promote the activity of luciferase (Figure 1D). A total of 202 CREs were predicted in the 2.5 kb sequence (Appendix A and Appendix A), including sex determination double sex element (DSX) (−1105 to −1093 bp) [39], elements of the early ecdysone-inducible gene Broad-complex (BRC) [40], and E74A (−106 to −92 bp) [41]. We also found some elements potentially binding to various homeodomain transcriptional factors: a homeobox transcription factor containing the POU domain, the CUT domain, and the Abd-B group (Appendix A and Figure 1E). In addition, a comparison of CREs in the *VgR* promoters of six insects revealed that multiple POU domain ventral veins lacking (VVL) elements were located in the six promoter regions of *VgR*, especially in Lepidoptera (Figure 1F and Appendix A). These findings led to the hypothesis that the POU domain family might play a crucial role in regulating *VgR* expression in insects.

### 3.2. Analysis of the Expression Pattern of POU-M2 and BmVgR in Ovary

To verify the effect of POU-M2 on *BmVgR* expression, we compared the expression patterns of *POU-M2* and *BmVgR* genes in the ovary at different developmental stages. Previous reports using reverse transcription PCR (RT-PCR) showed the expression of *BmVgR* mRNA was high in ovarian tissue at the wandering and pupal stages, when the translated VgR protein started to transport yolk protein for ovarian development [42]. Our qRT-PCR results revealed that the expression of *POU-M2* reached a peak at W2D (Figure 2A’). Similarly, *BmVgR* expression reached a peak at W3D (Figure 2A). At the same time, we found overexpression of POU-M2 promoted the transcription of endogenous *BmVgR* in BmNs cells (Figure 2B). These results indicated POU-M2 might activate the expression of *BmVgR* at the wandering stage.

### 3.3. POU-M2 Regulated BmVgR Expression by Binding Directly to the POU CREs of the BmVgR Promoter

To verify the effect of POU-M2 on the activity of the BmVgR promoter, differentially truncated BmVgR promoter vectors (p-VgRP-2.0k, p-VgRP-0.7k, and p-VgRP-0.1k) were co-transfected, along with the 1180-POU-M2 expression vector, containing the POU-M2 open reading frame into BmNs cells for assessment of luciferase activity. The BmVgR promoter activities were significantly enhanced by excess POU-M2 (Figure 3B). In addition, we wanted to verify if the POU-M2 protein could bind directly to the BmVgR promoter and enhance the promoter activity. We predicted twenty-one POU CREs in the 2.5 kb region upstream of BmVgR, based on the conserved POU element sequence (Figure 3A’). Among them, we randomly selected POU elements close to the BmVgR gene (POU CRE-1/2/3/4/5/6/7), which were also predicted by the JASPAR^2020^ program (Appendix A) for probe preparation (P1/P2/P3/P4/P5/P6/P7) (Figure 3A,C). The POU-M2 protein was expressed, purified, incubated with POU CRE probes in vitro, and detected by EMSA. All seven probes bound to the POU-M2 protein, and the binding was inhibited by competitive probes (Figure 3C,E). We only predicted a single POU CRE (POU CRE-1) within 0.1 kb of the BmVgR promoter (Appendix A), and the DLR assay showed that the promoter activity could not be enhanced by excess POU-M2 when this CRE element was mutated (Figure 3B). As shown in Figure 3D, when the concentration of probe P1 was increased, the binding signal was gradually enhanced, whereas with the addition of the competitive probe (Cold P1), the signal gradually weakened. The POU-M2/P1 binding band disappeared in the presence of a mutated probe (P1-M), and as the amount of anti-POU-M2 increased, the band became weaker (Figure 3D). When the nucleoproteins of POU-M2-overexpressing BmNs cells were used for the EMSA assay, similar results were obtained (Appendix A). Subsequently, POU-M2 and Myc antibodies (anti-POU-M2 and anti-Myc) were used to immunoprecipitate the myc-POU-M2 protein complex, which was overexpressed in BmNs cells after cross-linking with formaldehyde, and PCR amplification was performed on the precipitates. Both the precipitate captured by the anti-POU-M2 (Figure 4A,A’,A’’) and that captured by the anti-Myc (Figure 4B,B’) could be amplified by BmVgR promoter-specific primers. The fragments obtained were sequenced-verified (Figure 4C,C’). These experiments further demonstrated that POU-M2 bound to POU CREs in the BmVgR promoter and enhanced the activity of the BmVgR promoter.

### 3.4. Suppression of POU-M2 by RNA Interference Reduces the Expression of BmVgR and the Egg-laying Rate

Although *POU-M2* genes have been reported to regulate a cuticle protein gene (*BmWCP4*) [29] and a *BmVg* gene [31] in *B. mori* treated *POU-M2* dsRNA, the effect of *POU-M2* on the *BmVgR* gene has not been reported *in vivo*. So, we conducted RNA interference (RNAi) of *POU-M2* using siRNA. POU-M2-specific siRNA (Appendix A) was injected into the ninth abdominal segment at W0D and W1D (Figure 5A), and 24 hours later the ovary was dissected for RNA extraction. Oviduct development was severely blocked at P7D: the ovary was shorter (Figure 5B), and the egg-laying rate was lower than the controls (Figure 5C,C’). Moreover, the qRT-PCR analysis showed the expression of *POU-M2* was significantly reduced by *POU-M2*-specific siRNA injection (Figure 5D) and *BmVgR* upregulation was prevented (Figure 5D’). Further, BmVgR expression was decreased by POU-M2 suppression. Taken together, these results show that abnormal expression of *POU-M2* and *BmVgR* cause a significant impairment of ovary development.

## 4. Discussion

*VgR* is known to be highly conserved among insects and other species. Yet, at present, there are few studies on its transcriptional regulation. In our study, we found that many POU CREs were located in the *VgR* promoter region of six insect species representing Lepidoptera, Diptera, and Hymenoptera. Moreover, we verified that a POU domain transcription factor, POU-M2, directly bound to the regulatory region upstream of *BmVgR* and activated its transcription in *B. mori*. Analyses of expression patterns suggested that *POU-M2* was expressed initially in ovary at W2D, followed by *BmVgR*. Decreased *BmVgR* expression was observed in the ovary when POU-M2 expression was downregulated at the wandering stage of silkworm. There observations support the hypothesis that POU-M2 acts as a molecular switch to trigger the expression of *BmVgR* at the wandering stage.

A characteristic of POU-domain proteins is that they rely on their two specific subdomains to recognize diverse sets of DNA sequences. In general, POU domain transcription factors interact with promoter sequences based on the octamer consensus element ATGCAAAT [43,44]. However, in *D. melanogaster*, the POU domain transcription factor VVL has been shown to bind three distinct *VVL* recognition elements that are only weakly related to the octamer consensus motif and to each other [45,46]. In *B. mori*, POU-M1 has been reported to bind to the A/T-rich elements of the *POU-M1* gene, but cannot recognize any of the octamer-like sequences in the upstream region of the *POU-M1* promoter itself [47]. However, POU-M1 can recognize the octamer-like sequences in the upstream region of the *sericin-1* promoter, indicating that such a binding specificity is probably important for its functional interaction [48]. In 2012, Deng *et al*. reported that POUM2 could bind to the octamer-like sequences (CTTTACAT) of the silkworm wing disc cuticle protein (*BmWCP4*) gene [29], and in 2017, Lin *et al*. reported its binding sequence as the A/T-rich elements (TTATTAAA) upstream of the *BmVg* gene [31], which may be a binding site for POU_H_ subdomain. In other words, the DNA sequence recognition of POU domain proteins is flexible. In our case, POU-M2 can bind to the A/T-rich elements (P1/P5/P6) of the *BmVgR* gene with a lower affinity, and also bind to the octamer-like elements (P2/P3/P4/P7) with a higher affinity, indicating that multiple POU-M2-binding sites might be involved in the precise control of *BmVgR* gene transcription.

When POU-M2 expression was downregulated by siRNA injection, decreased *BmVgR* expression was observed in the ovary, together with a low egg-laying rate. Although the reduction of the *BmVgR* gene by dsRNA injection can lead to abnormal egg development and low egg-laying rate [42], we cannot rule out some indirect effects that could lead to the low egg-laying rate, such as the decrease of POU-M2, possibly accompanied by a drop in *BmVg* expression in the fat body [31], as well as by the inhibition of hormone synthesis [30]. In the future, knocking out or knocking down *POU-M2* only in the ovary through genetic approaches may be possible to further determine the effect of *POU-M2* on ovarian development. Nevertheless, the fact that *POU-M2* suppression caused *BmVgR* downregulation in the ovarian tissue is an important reason for the reduction of the egg-laying rate.

In this study, we verified that a conserved POU domain transcription factor, POU-M2, can upregulate *VgR* expression in *B. mori*. *VgR* regulation by POU domain transcription factors may be a common phenomenon in insects, as well as in other oviparous species. In the past three decades, POU domain transcription factors were identified in insects and many other oviparous species, such as the fishes *M. salmoides* and *Paralichthys olivaceus*, the amphibian *Xenopus laevis*, and the nematode *C. elegans* [7,49,50,51,52]. Notably, we found multiple POU CREs in the *VgR* promoters of six insect species representing three diverse taxonomic groups. It was also reported that POU CREs exist on the upstream regulatory region of the *VgR* gene in *M. salmoides* [7]. These clues suggested a possible regulatory role of POU factors in *VgR* transcription.

Although transcriptional regulation during insect oogenesis has been extensively studied [53], there is no direct evidence that *VgR* is regulated by POU domain transcription factors. The lack of reports about the involvement of POU domain factors in oogenesis may due to their role as vital regulators of insect metamorphosis, such as that their mutation or knockdown may result in embryonic lethality or abnormal metamorphosis, which occurs before egg development. When we injected the siRNA of POU-M2, we chose the ninth abdominal segment of the silkworm for siRNA injection, not the second thoracic segment for dsRNA injection, as described by Deng et al. [29]. Therefore, we observed a molting delay normal pupae, and arrested egg development. In the present studies in silkworm, we know that POU-M2, in addition to being essential for embryogenesis, silk gland development, and metamorphosis, plays an important role for ovarian development by regulating *BmVgR*.

## 5. Conclusions

We have found that conserved POU domain transcription factors directly bind to CREs in the *BmVgR* promoter, thus inducing *BmVgR* expression and activating oogenesis in *B. mori*. This method of regulation may be conserved in insects. These results suggest a new function of the conserved POU transcription factor family in *BmVgR* regulation, and expand the understanding of the POU transcription factor in insect *VgR* regulation.

## Figures and Tables

**Figure 1 genes-11-00394-f001:**
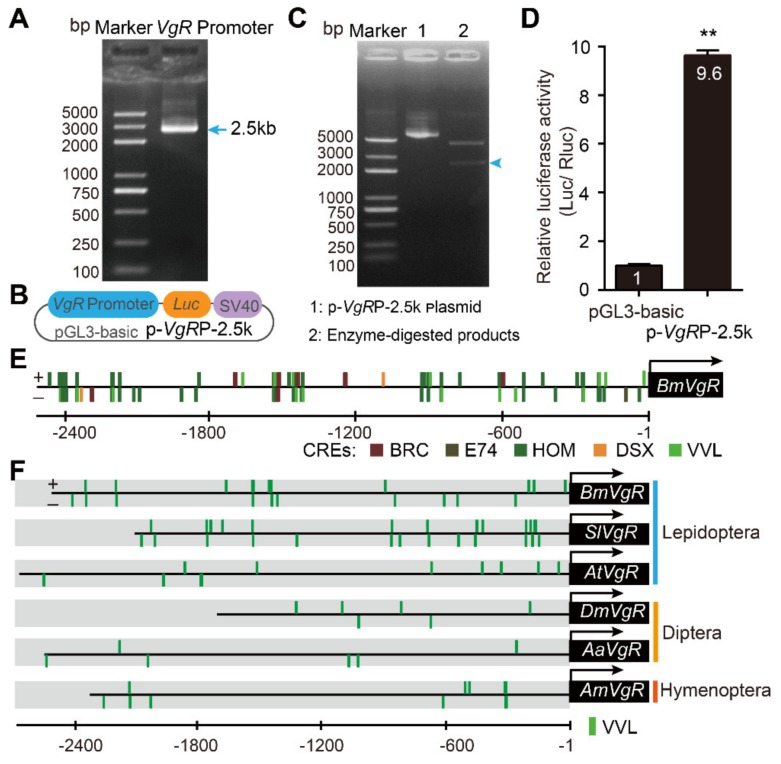
Analysis of the upstream regulatory region of the vitellogenin receptor (*VgR*) gene. (**A**): Clone of the upstream regulatory region of the *Bombyx mori* (*BmVgR*) gene. (**B**): Schematic diagram of the *BmVgR* promoter reporter vector. (**C**): Verification the *BmVgR* promoter reporter vector by enzyme digestion of Sma I and Hind III. (**D**): Activity verification of the *BmVgR* promoter by a dual luciferase reporter assay. The promoter activity is represented as fold-change compared to the control vector (pGL3-basic). (**E**): Diagram/Schematic of the cloned *BmVgR* promoter region, labeled with identified putative cis-response elements (CREs). Numbering of the upstream region sequence of the cloned *BmVgR* promoter is relative to the position of the translation start site (+1). Putative binding sites are: BRC, broad complex for ecdysone steroid response; E74, ecdysone induced protein E74A; HOM, homeodomain proteins; DSX, sex determination transcription factor doublesex; VVL, transcription factors with POU-domains. (**F**): Prediction of POU transcription factor cis-response elements (POU CREs) (VVL) in the *VgR* promoters of six insects. *Sl*, *Spodoptera litura*; *At, Amyelois transitella*; *Dm, Drosophila melanogaster*; *Aa, Aedes aegypti*; *Am, Apis mellifera*. The asterisks (**) represent significant differences (*p* < 0.01) detected using two-tailed Student’s *t* tests. Error bars indicate Standard Error of Mean (SEM) ranges (*n* = 3).

**Figure 2 genes-11-00394-f002:**
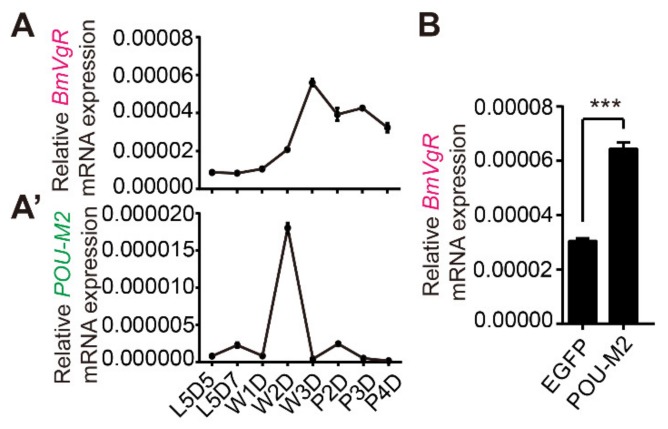
POU-M2 enhances the transcription of *BmVgR in vitro*. (**A**) and (**A’**): Correlation analysis of *POU-M2* and *BmVgR* expression, which is relative to the expression of the *eukaryotic translation initiation factor 4A* gene (*BmTIF4A*). Expression patterns of *BmVgR* (**A**) and *POU-M2* (**A’**) in the ovary at different stages. L5D5/7, day 5/7 of the 5th larval instar; W1D/W2D/W3D, days 1/2/3 of wandering; P2D/P3D/P4D, pupal days 2/3/4. (**B**): Effect of overexpression of POU-M2 on endogenous *BmVgR* expression in BmNs cells. The asterisks (***) represent significant differences (*p* < 0.001) detected using two-tailed Student’s *t* tests. Error bars indicate SEM ranges (*n* = 3).

**Figure 3 genes-11-00394-f003:**
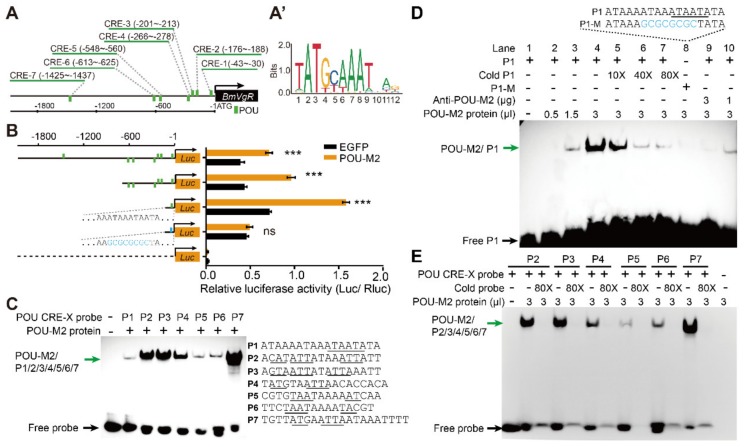
POU-M2 activates the expression of *BmVgR* by directly binding to its promoter. (**A**): Seven putative POU CREs in the *BmVgR* promoter region (Green boxes). (**A’**): Putative conserved sequences of POU elements from JASPAR ^2020^. (**B**): Effect of overexpression of POU-M2 on the activity of truncated and mutated *BmVgR* promoters (**C**): EMSA assays performed to evaluate the binding of the seven POU element probes (P1/P2/P3/P4/P5/P6/P7) and POU-M2 protein. Probe sequences are shown on the right and possible core binding sites of POU elements are underlined. (**D**): Binding of POU CRE-1 to the POU-M2 protein with cold probe, mutant probe, and anti-POU-M2 antibodies. Light blue represents the mutated sequence of POU CRE-1. (**E**): Competitive binding of POU CRE-2/3/4/5/6/7 to the POU-M2 protein with an 80-fold cold probe excess. All data represent three biological replicates with three technical replicates; error bars indicate SEM ranges (*n* = 3). *** *p* < 0.001 (Student’s *t* test). ns: not significant.

**Figure 4 genes-11-00394-f004:**
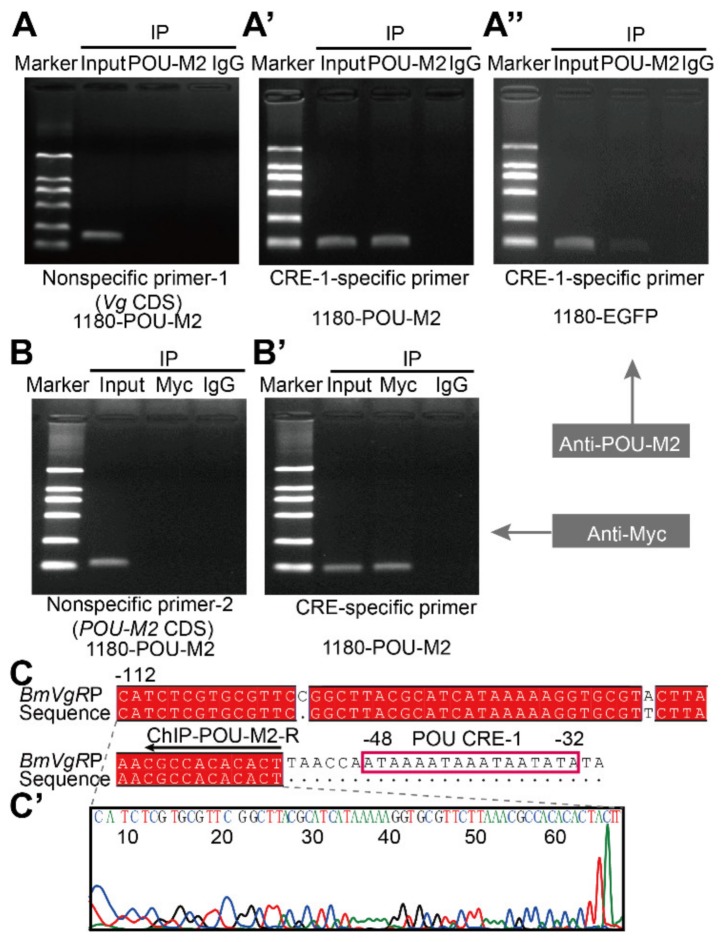
ChIP assays to evaluate the binding of POU-M2 to overlapping CREs in the *BmVgR* promoter. (**A**) and (**B**): PCR results (nonspecific primers) from the ChIP assays in BmNs cells overexpressing Myc-tagged POU-M2. (**A’**), (**A’’**), and (**B’**): PCR results (POU CRE-specific primers) from the ChIP assays in BmNs cells overexpressing Myc-tagged POU-M2 and EGFP. (**C**) and (**C’**): Sequences of the specific PCR band from the ChIP assays.

**Figure 5 genes-11-00394-f005:**
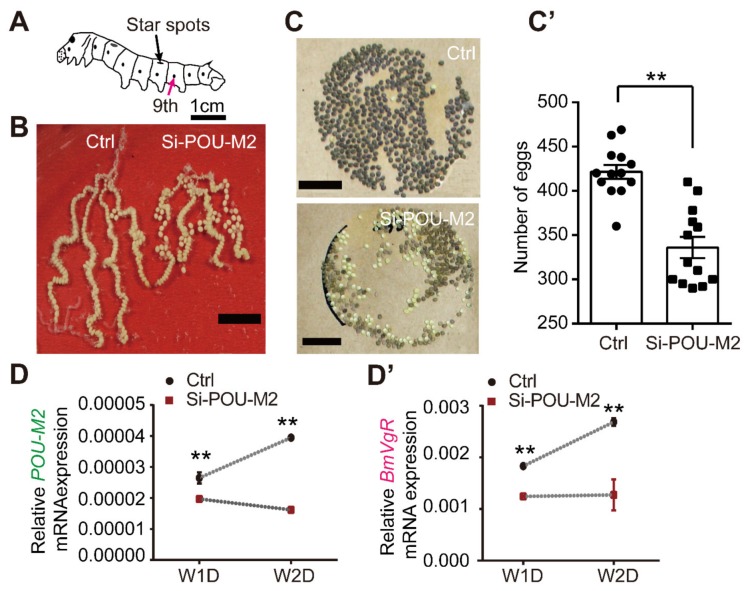
*POU-M2* suppression by RNAi caused a decrease in the egg-laying rate. (**A**): Schematic diagram of the injection site (magenta arrow) of silkworms at W0D and W1D. (**B**): Development of the ovariole at P7D after *POU-M2* downregulation. (**C**) and (**C’**): Visual demonstration and statistics of egg laying in the RNAi group and the control group (*n* = 11). (**D**) and (**D’**): *POU-M2* and *BmVgR* expression level after injection of *POU-M2*-specific siRNA. All data represent three biological replicates with three technical replicates, error bars indicate SEM ranges. ** *p* < 0.001 (Student’s *t* test).

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
