# Peer review of "The POU Transcription Factor POU-M2 Regulates Vitellogenin Receptor Gene Expression in the Silkworm, Bombyx mori"

_genes, 2020, doi:10.3390/genes11040394_

Round 1

Reviewer 1 Report

     A well-written manuscript detailing POU-binding sequences controlling expression of the vitellogenin receptor in the silkworm.  

     I enjoyed reading the article.  The presentation was mostly clearly and the science laid out in an orderly fashion.  I found a few things that I would recommend the authors change to strengthen the clarity of the manuscript. Hopefully the authors will find these items helpful:

Line 51 – Add “to” between “found” and “upregulate” and “the” between “in” and “ovary”.

Line 53 – species is not capitalized, only the genus.

Line 55 – is “choriongenesis” a word?  Choriogenesis, perhaps?

Line 64 – “POU specific” becomes “POU-specific”

Line 75 “steroid ogenic enzyme genes”? “steroidogenic enzyme genes?

Lines 86/87 – “…at a humidity level appropriate for hatching”.  This isn’t helpful.  Be helpful by stating the humidity level.

Line 121 – “mensuration” should most likely be “measurement”

Line 140 – slow cooling to any particular temperature (25C) prior to the EMSA?

Line 147 –“ultraviolet” should be followed by the word “light”.

The role of Ventral Veins Lacking (VVL) as an important POU response element needs to be explained in the introduction otherwise Figure 1 will be less intelligible. Perhaps some of the information shared in lines 294-299 can be incorporated into the introduction as well?

Line 203-  heading 3.3 “Analyze the expression pattern…” should read:  “Analysis of the expression pattern…”

Figure 2 – Y axis labels – relative expression levels.  Relative to BmTIF4A? I know it is mentioned in the M&M but it may be worth putting the control in the legend of this figure.

Line 221 – “differentally” = “differentially”

Line 224 – “additon” = “addition”

Line 225 – “anhance” = “enhance”

Line 295 – This sentence appears incomplete.

Line 316 – “further” should read “future”

Line 317 – sentence is incomplete “to further observation”?

Line 324  - “Salmoides” should be lower case.

Lines 328-332 – Good summary sentences but require a grammar check.

Author Response

Dear Reviewer:

We would like to express our great appreciation to you for your professional comments and suggestions regarding our study. The comments are valuable and have been very helpful for revising and improving our manuscript. We have carefully improved the manuscript according to your comments. The revised portions of the manuscript text are marked in red. Our specific responses to the comments please see the attachment.

Reviewer 2 Report

Shen et al., herein, concluded that POU-homeodomain transcription factor, POU-M2, regulates expression of the VgR gene in oogenesis of Bombyx mori.  They speculate that the role of POU-homeodomain protein in oogenesis may be universal in insects.

     The conclusion was obtained mainly based on the results from 1) functional analyses of cloned VgR promoter in cellulo (POU-M2-overexpressing BmN cells), 2) ChIP experiments using the same cells, 3) EMSA using recombinant POU-M2 protein, and 4) RNA interference experiments using Bombyx larvae with POU-M2 siRNA.  The results of these experiments in cellulo and in vitro are clear.  The expression kinetics of POU-M2 and VgR genes in the ovary analyzed by qRT-PCR strengthen their conclusion.  However, this research is premature to lead the conclusion.  Information about roles of POU-M2 protein and CRE1 (cis -element in the promoter) of the VgR gene in the ovary is insufficient. Though RNAi experiments were carried out, assessment of the results is difficult, because indirect effects might lead the results as the authors described in Discussion.  Furthermore, many research groups suggest that the mechanism of RNAi does not work well with exogenous siRNA (by the injection to larvae) in the silkworm.

Major points

     Please supply more information about POU-M2 and VgR proteins and their genes in the ovary of the silkworm, and answer following questions.

1) The ovary (or ovariole) contains different cells such as eggs, nurse cells, follicle cells and sheath cells.  Does POU-M2 protein exist in the VgR expressing cells on the appropriate time?  The data by immunohistochemistry and in situ hybridization experiments are necessary.

2) Does CRE1 work as a promoter element in vivo? Examine activity of the short promoter of VgR gene in transgenic silkworm.  If preparation of the transgenic silkworm is difficult, it may be possible to use transgenic fly, because the authors suggest that the function of POU-homeodomain protein is universal in insects.

3) Does POU-M2 bind to CRE1 in vivo?  The data of ChIP experiment using ovary (or ovariole) is necessary.

4) Is there any reason to exclude the possibility that other homeodomain proteins than POU-M2 bind to CRE1 and activate the VgR gene?  CRE1 and other elements in the VgR promoter are AT-rich and contain universal homeodomain-binding sequences.

Minor point

Lines 195-197: “Numbering of the upstream region sequence of the cloned BmVgR promoter is relative to the position of the transcription start site (+1)” for Fig.1.  No description for Fig. 3, but “ATG” is shown in Fig. 3A.  In Fig.S1, “Nucleotides are numbered relative to the translation start site (+1)”.  Are these descriptions correct?  Had the transcription start site of VgR gene been confirmed?

Author Response

Dear Reviewer:

We would like to express our great appreciation to you for the comments and suggestions regarding our study. The comments are valuable and have been very helpful for revising and improving our manuscript, as well as providing important guidance in our studies. We have examined the comments carefully and have made corrections to the manuscript accordingly, which we hope will meet with the Reviewer’s approval. The revised portions are marked in blue in the revised version. The main corrections in the manuscript and our detailed responses to the comments are in the attachment.

Round 2

Reviewer 2 Report

Whether RNAi works well in the silkworm or not, secondary effects are possible in this case as the authors described.  Therefore, data from in vivo experiments is still insufficient to lead the conclusion.  Either the data of 1) immunohistochemistry or 2) ChIP using the ovary is necessary at least. 

However, I understand situation of the authors that their Lab. is being closed because of COVID.  If the authors want to publish the manuscript with present form without additional in vivo data, I suggest to change the title to euphemistic one.

Author Response

Dear Reviewer:

We would like to express our great appreciation to you for your understanding and valuable suggestions. we have changed the title “The POU transcription factor POU-M2 controls silkworm oogenesis by regulating vitellogenin receptor gene transcription” to “The POU transcription factor POU-M2 regulates vitellogenin receptor gene expression in the silkworm, Bombyx mori” according to your suggestions. About the part of works as you mentioned will be continue.